A novel approach for credit card fraud transaction detection using deep reinforcement learning scheme

http://orcid.org/0000-0002-5510-6078 Qayoom Abdul 1 2
http://orcid.org/0000-0002-2447-9816 Khuhro Mansoor Ahmed 3
Kumar Kamlesh 4
http://orcid.org/0000-0002-0331-6276 Waqas Muhammad 5
Saeed Umair 6
ur Rehman Shafiq 7
http://orcid.org/0009-0001-4576-830X Wu Yadong 1 8 wuyadong@suse.edu.cn
http://orcid.org/0000-0002-7047-8726 Wang Song 1
1 School of Computer Science and Technology, Southwest University of Science and Technology , Mianyang, Sichuan , China
2 Department of Computer Science, Lasbela University of Agriculture, Water and Marine Science , Uthal, Lasbela, Balochistan , Pakistan
3 Department of Artificial Intelligence and Mathematical Sciences, Sindh Madressa-tul-Islam University , Aiwan-e-Tijarat Road, Karachi, Sindh , Pakistan
4 Department of Software Engineering, Sindh Madressa-tul-Islam University , Aiwan-e-Tijarat Road, Karachi, Sindh , Pakistan
5 School of Software Engineering, University of Electronic Science and Technology of China , Chengdu, Sichuan , China
6 Department of Computer Science, Bahria University , Islamabad , Pakistan
7 Department of Computing and Information Technology, Mir Chakar Khan Rind University of Technology , Dera Ghazi Khan, Punjab , Pakistan
8 School of Computer Science and Engineering, Sichuan University of Science and Engineering , Zigong, Sichuan , China
Alarcon-Aquino Vicente
Electronic publication date: 2024 Apr 29
Publication date: 2024
Volume: 10
Electronic Location ID: e1998
Received 2023 Dec 30; Accepted 2024 Mar 27
Copyright: © 2024 Qayoom et al.
Copyright year: 2024
Copyright holder: Qayoom et al.
License: This is an open access article distributed under the terms of the Creative Commons Attribution License, which permits unrestricted use, distribution, reproduction and adaptation in any medium and for any purpose provided that it is properly attributed. For attribution, the original author(s), title, publication source (PeerJ Computer Science) and either DOI or URL of the article must be cited.
License URL: https://creativecommons.org/licenses/by/4.0/

Keywords: Credit card fraud transaction, Classification, Deep learning, Deep reinforcement learning, Real time detection, Business intelligence

Funding: The authors received no funding for this work.

==============================
Online transactions are still the backbone of the financial industry worldwide today. Millions of consumers use credit cards for their daily transactions, which has led to an exponential rise in credit card fraud. Over time, many variations and schemes of fraudulent transactions have been reported. Nevertheless, it remains a difficult task to detect credit card fraud in real-time. It can be assumed that each person has a unique transaction pattern that may change over time. The work in this article aims to (1) understand how deep reinforcement learning can play an important role in detecting credit card fraud with changing human patterns, and (2) develop a solution architecture for real-time fraud detection. Our proposed model utilizes the Deep Q network for real-time detection. The Kaggle dataset available online was used to train and test the model. As a result, a validation performance of 97.10% was achieved with the proposed deep learning component. In addition, the reinforcement learning component has a learning rate of 80%. The proposed model was able to learn patterns autonomously based on previous events. It adapts to the pattern changes over time and can take them into account without further manual training.

Introduction

With advancements in computing technology, Internet users can now upload, retrieve, and process huge volumes of data across remote locations via high-speed linked networks. There are various high-end applications of the Internet such as voice email, searching for patterns from repositories, navigating self-driving cars, performing financial transactions, and high bandwidth streaming services. However, increased usage of technology has gained the attention of attackers in various domains like intrusion in smart devices including credit card fraud, click fraud, procurement fraud, and identity theft. Fraud is a general term that can be defined as deceiving someone by stealing sensitive information to harm them financially or degrade their reputation. Furthermore, innovation in e-commerce has also attracted users to shift to online platforms for the purchase of goods and services that require a credit/debit card. After the COVID-19 pandemic, online shopping has gained popularity and still remains trending. On the other hand, this imposes high responsibility on financial institutions including banks and other online service providers to secure their clients’ credentials from cyber attackers for the potential transactions. In recent years, credit or debit card holders have been the victim of several notable crimes in fraudulent transactions or theft. Portions of this text were previously published as part of a preprint (Qayoom et al., 2023).

Credit card fraud involves accessing credit card information illegally on behalf of legitimate card owners either by physical or electronic means (Adewumi & Akinyelu, 2017). The figures provided in 2014 reported that over 490 billion dollars in fraudulent transactions were reported using credit cards (Dai et al., 2016). Contrary, to the high rate of legitimate transactions there are billions of dollars of fraudulent transactions occurring around the world every second. In financial industries such as banks, credit card fraud detection system aims to decide whether a transaction is based on historical data or not (Benchaji et al., 2021). Therefore, the detection of fraudulent transactions has paramount importance in online platforms. To optimize their performance, various machine learning and data mining techniques have been applied for the detection of such fraudulent transactions (Tanouz et al., 2021; Rout, 2021) but due to the lack of data availability and confidentiality issues, this area is still evolving with the latest research.

However, current research relies heavily on artificial intelligence to build more robust intelligent models that might provide high accuracy to analyze massive datasets. Among such methods, deep learning models provide remarkable results in various applications. Some notable areas are analysis of medical images, object recognition, natural language processing, genome structural analysis in Bio-informatics, and detection of fraudulent transactions. Many learning methods have been employed by researchers including deep learning (DL), reinforcement learning (RL), and deep reinforcement learning (DRL) (Mnih et al., 2015).

Therefore, decisions made with machine learning methods are based on the known features of the training data, while a criterion composed of loss, reward, and regret is usually optimized by the learning algorithm for training and test data (Singh et al., 2022). The difference between financial decision problems and the classical stochastic control approach is that the former is modeled with stochastic processes and control methods, while the latter deals with the agents acting in a system that can make optimal decisions by learning through repeated experiences by interacting with the system.

The work presented in “Related Work” provides the performance evaluation based on machine learning and deep learning algorithms. However, there are some research gaps and limitations where the evaluation metrics indicate poor model performance, and their reproducibility is limited by the lack of a feature selection procedure. Consequently, the lack of feature selection affects performance of the proposed models and the computational cost may limit the model’s applicability in practical scenarios. All these critical factors have motivated this research. In particular, financial security requires the detection of credit card fraud, and the use of cutting-edge technologies such as Deep Q Networks (DQN) has transformed the effectiveness of these systems. Using this method DQN can detect complex patterns and anomalies associated with fraudulent activity by training the model on a 30-day dataset of historical transaction data. DQN uses a reinforcement learning framework that, unlike traditional machine learning methods, allows it to dynamically update knowledge based on ongoing transactions and make predictions in real time. This flexibility ensures that fraud detection system is more robust and responsive. The DQN model improves accuracy by up to 97.10% over the previous 30 transactions Z-score for fraud detection by leveraging temporal patterns in the data to identify subtle patterns that previous models may have missed.

The main objective of this research is to detect credit card fraudulent transactions in real-time using deep reinforcement learning. The article has been arranged as follows: “Related Work” provides related work in credit card fraud detection. “Proposed Methodology” gives the proposed methodology and discusses features in the dataset. “Model Evaluation” provides model evaluation. Finally, “Conclusion” presents the conclusion.

Related work

Credit card fraud is a serious and growing problem and occurs in the form of skimming, counterfeiting, and phishing operations, which collectively result in billions of dollars in financial losses to consumers and businesses each year. The ongoing and evolving aspects of fraud require additional precautions, even with concerted efforts by banks and merchants to reduce these risks. Because credit cards are used for hundreds of transactions every day, prevention and detection methods need to be improved (Barker, D’amato & Sheridon, 2008). A wide range of available machine learning algorithms based on supervised learning, unsupervised learning, deep learning algorithms and ensemble learning algorithms for anomaly detection and detection of credit card fraudulent transactions in real-time have been used in different research works. Randhawa et al. (2018), used twelve different algorithms on a publicly available world financial institution dataset. They used the Matthews correlation coefficient (MCC) score (Powers, 2020) and Adaboost algorithm with majority voting, which has been integrated with these aforementioned standard algorithms and applied to the same public and financial institutions dataset. The combination of standard and hybrid models provided the 0.823 MCC score by majority voting on the publicly available datasets. When these methods were applied together to the financial institution dataset, they achieved a perfect 1 MCC score by Adaboost and majority voting methods. Furthermore, to verify stability of the model, 10% and 30% noise were added to the dataset. It achieved a 0.942 MCC score in presence of 30% noise in the dataset.

Unfortunately, the increase in credit card fraud in recent years is directly related to the increase in credit card transactions, resulting in significant losses for both consumers and merchants. To address this growing problem, three different fraud detection techniques have been proposed. To classify transactions as legitimate or fraudulent, a clustering technique is first used to analyze data clusters based on parameter values. Secondly, the probability density of the past behavior of a credit card user is captured using a Gaussian mixture model, which helps determine the likelihood of any deviations from the typical patterns of the user. Finally, Bayesian networks are used to illustrate statistical relationships between a specific user and other fraud scenarios. The goal of these approaches is to improve the overall accuracy of credit card fraud detection by gaining an accurate overview of the problem and understanding the important characteristics that emerge from the implementation of each method (Dheepa & Dhanapal, 2009). Alfaiz & Fati (2022) provided an experimental study for the detection of credit card fraud that is based on two stages. The output generated by the first stage is used as an input for the second stage. They used real-world European datasets on credit card fraud detection, which were obtained from Kaggle (Lebichot et al., 2020; Carcillo et al., 2021; Dal Pozzolo et al., 2017, 2014, 2015). They implemented nine machine learning algorithms on available datasets for the detection of fraudulent transactions and generated an evaluation matrix by stratified K-fold cross-validation in the first stage. As a result, three best performance algorithms have been selected for the second stage. Then, they applied 19 re-sampling techniques (Alfaiz & Fati, 2022) on the same dataset and then tested the best three algorithms one by one in the second stage. They considered the All K-nearest neighbors (AllKNN) under-sampling technique along with CatBoost (AllKNN-CatBoost) as the best-proposed model, which provided an AUC value of 97.94%, a recall value of 95.91%, and an F1-score value of 87.40%. Benchaji et al. (2021) performed experimental research over two datasets. The first dataset was obtained from Kaggle which has a real-world European dataset on credit card fraud detection within 2 days (Lebichot et al., 2020; Carcillo et al., 2021; Dal Pozzolo et al., 2017, 2014, 2015). They used simulation software named Banksim for the generation of the second synthetic dataset. A total of 594,643 transactions were made throughout 180 simulated days in the second dataset (Vaughan, 2020). By using multiple dimensionality reduction, feature selection, and extraction algorithms like principal component analysis (PCA) (Jolliffe & Cadima, 2016), uniform manifold approximation and projection (UMAP) (Becht et al., 2019; McInnes, Healy & Melville, 2018), and t-distributed stochastic neighbor embedding (t-SNE) (Linderman et al., 2019; Van der Maaten & Hinton, 2008) for the optimization of the learning classifiers. They applied the Synthetic Minority Oversampling Technique (SMOTE) (Chawla et al., 2002; Kumari & Mishra, 2019) to improve the learning rate and better handling of the imbalanced dataset. They implemented the attention mechanism upon LSTM recurrent networks which improved the performance of their model for the detection of fraudulent transactions and provided 96.72% accuracy, 98.85% precision, and 91.91% recall upon dataset 1 while 97.48% accuracy, 97.69% precision and 97.22% recall upon dataset 2. Braun et al. (2017), proposed a fraudulent transaction detection approach based on payment patterns, which were done on suspicious points of sale. With the help of pattern matching, the learning algorithm detected 50% of fraudulent transactions at the point of sale. The results were verified by World Line (industrial partner). The dataset was collected in September 2013 from Europe. They implemented a random forest algorithm and the model achieved 99.9% of accuracy with 33.3% of precision, 38.1% of recall, and AUC of 97.1%. Hemdan & Manjaiah (2022), provided a comparison analysis among deep learning and machine learning algorithms for anomaly detection in credit card fraud. Similarly, Fu et al. (2016), utilized a convolution neural network (CNN) (Wiatowski & Bölcskei, 2017) to detect a hidden pattern of fraudulent transactions in credit cards. The model was evaluated over a dataset of a commercial bank which contained 260 million transactions. By using the CNN model, they transformed transaction data features into a feature matrix. They applied a random under-sampling technique to handle imbalanced data, which is applied over the majority class due to which valuable information can be lost. To avoid this problem, they used cost-based sampling technique for the detection of fraudulent transactions. For feature generation, some methods have been described in Bhattacharyya et al. (2011), Khandani, Kim & Lo (2010), Ravisankar et al. (2011) and Van Vlasselaer et al. (2015) but they could not detect complex patterns, Fu et al. (2016) introduced a novel strategy for mining complicated latent fraud patterns named as trading entropy. They concluded that data with trading entropy perform better than the ones without entropy. Zakaryazad & Duman (2016) deployed an artificial neural network (ANN) (Hopfield, 1988) and represented variants of it like Single-layer perceptron (SLP) and multi-layer perceptron (MLP). In their article, MLP has been implemented over two real-time fraud detection datasets and a UCI repository benchmark dataset. The proposed model contained three layers including the Input layer, hidden layer, and output layer. For better performance of the model, some bias has been added in each neuron so that it can predict the target exactly. They considered cost-sensitive learning and designed the profit-based ANN model in such a way that the model will give a penalty at every misclassification and will grant the profit at every correct classification, the focus of their article is to maximize the profit. Based on weight, the learning ability of ANN has been increased by assigning more weight to the most costly instances. They compared the model with different eleven classifiers, in which ANN performed well in terms of accuracy and true positive rate. However, naïve Bayes outperforms in saving when the threshold is set to 0.5.

Williams, Lewis-Parks & Wolf (2023) found that many demographic, psychological, and financial characteristics make people more likely to become victims of credit card fraud. The study shows that there are specific characteristics within these three groups that may be statistically associated with a higher risk of becoming a victim of credit card fraud. The study aims to increase our understanding of the variables that contribute to vulnerability so that we can develop more targeted prevention and mitigation measures by examining the relationships between these variables and fraud incidents. Bolton & Hand (2002) in their review article suggested that effective statistical and machine learning methods for fraud detection, have played a key role in identifying various fraudulent behaviors including computer intrusions, e-commerce credit card fraud, money laundering, and telecommunication fraud. This review article describes the various applications of statistical fraud detection techniques as well as the other sensitive areas where these technologies are useful. These technologies demonstrate their adaptability and effectiveness in the evolving context of fraudulent activity across many financial industries and play a crucial role in protecting against financial crime and security breaches. Beigi & Amin Naseri (2020) assessed the issue of imbalanced class in the dataset for credit card fraud detection using statistical and data mining techniques. The base learner of the Adaboost algorithm was a cost-sensitive C4.5 tree and this method was applied to a large dataset from a well-known Brazilian bank. The authors evaluated the effectiveness of the classifiers using a measure of the cost of misclassification. The results showed that the proposed method reduced the cost of classification errors by at least 14.62% compared to alternative strategies such as NN, DT, AIS, NB, and BN. It is worth noting that the proposed method produced results with a sensitivity of 67.52% and an accuracy of 96.59%. Dai et al. (2016) mentioned a general four-layer framework which includes a streaming detection layer, key-value sharing layer, batch training layer, and distributed storage layer. The objective of the article is to detect online credit card fraud, because of the tremendous amount of transactions it was implemented on big data technology like Hadoop, Spark, Storm, and HBase. In their article, they created a simulated transaction dataset by Markov a modulated Poisson process model (Panigrahi et al., 2009). Hidden Markov model (HMM) (Iyer et al., 2011), DBSCAN (Panigrahi et al., 2009) unsupervised algorithms, and logistic regression (LR) (Yeh & Lien, 2009) supervised algorithms have been executed. In their work, they were not only concerned with the accuracy but the detection time matter as well. The results showed that in the training model with an explicit filter, Spark outperformed Hadoop while in the training model with a quick filter, Hadoop outperformed Spark.

Tanouz et al. (2021) applied distinct machine learning (ML) algorithms like LR, naïve base (NB), decision tree (DT), and random forest (RF) on the same Kaggle dataset and got the maximum accuracy by RF, which is 96.77%. Sailusha et al. (2020) compared the AUC between two ML algorithms RF and Adaboost (AB), in which RF presented 94.29% AUC while AB presented 96.88% AUC. Raghavan & El Gayar (2019) implemented the various ML and DL algorithms including restricted Boltzmann machine (RBM), autoencoders (AE), RF, CNN, SVM, KNN, and ensemble (CNN, SVM, KNN), in which RBM performed the best and got 91.09% AUC. Esenogho et al. (2022) implemented the various algorithms such as SVM, MLP, DT, AB, LSTM and LSTM ensemble (proposed) on the Kaggle dataset, in which their proposed method acquired the highest AUC of 89%. Khalid et al. (2024) implemented the various algorithms like SVM, KNN, RF, LR, bagging, and boosting, proposed method 1 and 2 (PM1, PM2) on the Kaggle dataset, in which their PM1 and PM2 gained the accuracy of 93.68% and 94.73%.

Furthermore, Table 1 highlights the comparative analysis between available literature reviews on the same Kaggle dataset (Lebichot et al., 2020; Carcillo et al., 2021; Dal Pozzolo et al., 2017, 2014, 2015).

Table 1 Comparative analysis between available methods on Kaggle dataset.

Authors	Year	Method	AUC	
Braun et al.	2017	Random forest	97.10%	
Randhawa et al.	2018	MCC score by Adaboost	94.20%	
Raghavan & El Gayar	2019	Restricted Boltzmann machine	91.09%	
Sailusha et al.	2020	Adaboost	96.88%	
Tanouz et al.	2021	Random forest	96.77%	
Benchaji et al.	2021	LSTM	96.72%	
Esenogho et al.	2022	LSTM ensemble (proposed)	89.00%	
Khalid et al.	2024	Proposed method 2 (Ensemble)	94.73%	
Our proposed	Present	Proposed model (using DQN)	97.10%	

Proposed methodology

In this section, Deep Q-network is briefly discussed to explain the credit card fraud transaction dataset and pre-processing step. In the next step, proposed model and simulation are elaborated in detail. In the end, we provide the proposed system design and its architecture.

Deep Q network

Deep Q network (DQN) is one type of reinforcement learning (RL) that is mainly used to guide an artificial agent to select optimal action for performing a certain state without any interaction. Deep Q Network (DQN) combines deep neural networks with the Q-learning algorithm. The Q-learning algorithm is a well-known method in reinforcement learning that learns an optimal action-value function Q(s, a), which gives the expected cumulative reward for taking action ‘a’ in state ‘s’ and following an optimal policy thereafter.

The Q-value function in DQN is approximated using a deep neural network and the states of the environment serve as the inputs to the network and the Q-values of all possible actions as output. A variant of the Q-learning algorithm is used to train the neural network and the network weights are adjusted to minimize the difference between the target and expected Q values. In DQN, a deep neural network is used to approximate the Q-value function, where the inputs to the network are the state of the environment and the outputs are the Q-values of each possible action. The neural network is trained using a variant of the Q-learning algorithm, where the network’s weights are updated to minimize the difference between the predicted Q-values and the target Q-values. In addition, the Markov decision process (MDP) describes the state of the environment to an agent. In an environment, S0, S1, and S2 provide different states in which an agent executes actions (A) in an environment. An agent gets a reward or penalty based upon an action achieved in the current state, π:S→A. When an agent receives the reward, it moves to the next state. However, an agent receives feedback from the reward or penalty between state transitions and designs a strategy for future actions. For reinforcement learning, a mathematical model is built on the Markov decision process (MDP), and it is denoted by the set of states (S), which are a set of actions (A), probability function (P), discount factor ( γ), value (v) and reward function (R). Furthermore, initial state S0 comprises of sequence of actions that lead to the dynamics of an MDP. The following Eq. (1) provides the sequence of states and actions.

(1) S0→a0⁡S1→a1⁡S2→a2⁡…

(2) Qπ(s,a)=E[Σk=0∞γkR(Sk,π(sk))|π,s0=s,a0=a]

(3) Q*~(s,a)=Et−Psa[R(s,a)+γmaxbεA Q*~(t,b)]

(4) Qt+1~(s,a)=Et−Psa[R(s,a)+γmaxbεAQt~(t,b)]

(5) Li(Θi)=E(s,a)−ρ[(Et−Psa[R(t,a)+γmaxbεAQDNNθ−1(t,b)]−QDNNθ−1(s,a))2].

An optimal policy for MDP is chosen with a Q-learning search algorithm to find future rewards from the current state S. As shown in Eq. (2), an average discounted sum of rewards R is defined by Q-function. From the current state s, all possible paths represent the expected benefits-gain in the future. Further, when the mapping size of |S|∨|A| gets larger it yields more complexity in processing as mentioned in Eq. (3), and the optimal value of Q-function is derived through Eq. (4). To overcome this problem, Google DeepMind proposed an efficient supervised learning solution for Q-function by using deep neural networks (DNNs). The comprehensive overview of DNNs for the Q function has been provided in Mnih et al. (2015, 2013). The mathematical formulas of computation of DNN are given in Eqs. (5) and (6), where Θi indicates the weights of DNNs and Li( Θi) indicates the loss function to be minimized at ith iteration.

(6) Li(Θi)=E(s,a)−ρ[(Yi(s,a)−QDNNθ−1(s,a))2].

Furthermore, the loss function cannot be minimized equal to zero, there always exists a small error value.

Dataset

The dataset used in this research has been taken from the Kaggle dataset repository (Carcillo et al., 2021; Chawla et al., 2002; Dai et al., 2016; Dal Pozzolo et al., 2017, 2015). It consists of credit card transactions of European card holders. These transactions occurred during two days in which 492 fraudulent transactions were performed out of 284,807 transactions. The dataset is not normally distributed and highly imbalanced. The fraudulent transaction is about 0.172% of total transactions. The dataset contains 31 feature sets and one target column and all the feature sets contain numeric values. Due to confidentiality issues, most of the feature sets have been renamed and V1 to V28 columns are obtained by using principle component analysis. Only two columns “Time” and “Amount” have been mentioned with real name. The time feature contains the elapsed time between every transaction and the initial transaction. The feature amount contains the transaction amount and the target class is the binary representation, where 1 for fraud and 0 for benign. It can be observed from Fig. 1 that only Time, Amount, and Class are closely related. All the features are numeric so no conversion is required for any feature set. There are no missing values in the dataset. No pre-processing has been performed on the dataset and features have been obtained from PCA.

Figure 1 Features correlation heatmap.

Proposed system architecture

In this section, the main architecture of the proposed server-side application has been discussed in which the system is implemented in C# C++, and Python. The communication layer has been written using Boost library and OpenAI python is used for deep reinforcement learning algorithm. The C++ wrapper layer has been written over Python implementation. For data streaming, Apache Kafka has been used in the system prototype. Figure 2, represents the block diagram of system architecture. However, socket communication between the client and server is achieved through the communication manager. The communication manager contains the protocol manager. In the protocol manager module, protocol parsing and building have been implemented accordingly. To load and store the weights of the proposed DQN model along with data, a cache manager and data access have been designed. The process manager is one of the core components of the proposed system, and this is responsible for processing incoming messages and generating responses to send back to the client. The process manager includes ProcessManager.cpp, CommunicationManager.cpp, Message.cpp, ConfigMnager.cpp, CacheManager.cpp and DeepLearningProcess.cpp which are the core classes in system architecture. As shown in Fig. 2, the ProcessManager class contains the instances of most of the core classes. The Algorithm 1 shows the logic to process any incoming message and save it in multi-map.

Figure 2 Block diagram of proposed system architecture.

Algorithm 1 Process incoming message and save tags & values in a multimap.

1: Input: Multimap tagValueMultimap	
2: Output: None	
3: Function ProcessIncomingMessage	
4:   incomingMessage← GenerateIncomingMessage	
5:   parsedMessage← ParseMessage(incomingMessage)	
6:   tags←parsedMessage.tags	
7:  values←parsedMessage.values	
8:  for each tag, value in Zip(tags, values) do	
9:   PutInMultimap(tagValueMultimap, tag, value)	
10:  end for	
11:  return “tagValueMultimap”	
12: end function	
13: Function GenerateIncomingMessage	
14:  // Simulate the generation of an incoming message	
15:  return “tag1:value1,tag2:value2,tag3:value3”	
16: end function	
17: Function ParseMessage(message)	
18:  // Parse the incoming message and extract tags and values	
19:   tagsAndValues← Split(Message)message	
20:   tags←[]	
21:   values←[]	
22:  for each tagValue in tagsAndValues do	
23:    tag,value← ExtractTagAndValue(tagValue)	
24:   Append(tags, tag)	
25:   Append(values, value)	
26:  end for	
27:  return {“tags”: tags, “values”: values}	
28: end function	
29: function SplitMessage(message)	
30:  // Split the message based on a delimiter (e.g., comma)	
31:  return Split(message, ‘,’)	
32: end function	
33: function ExtractTagAndValue(tagValue)	
34:  // Extract the tag and value from a key-value pair	
35:   tag,value← Split(tagValue, ‘:’)	
36:  return tag, value	
37: end function	
38: Function PutInMultimap(multimap, tag, value)	
39:  // Put the tag and value in the multimap	
40:  Put(multimap, tag, value)	
41: end function	

Environment setup

An environment has been designed with input data and a history range that provides previous transactions in specific periods. In this research, the default history range is 30 to 60 previous transactions. Two processes have been defined in the environment, one is the “Reset” process and the other is the “Step” process. The first one re-initializes all parameters used in the environment such as z-score re-initialization to 0. The second one returns the feedback to the agent based on its provided action. In this process, the z-score of the previous 30 transactions has been calculated. However, if the calculated z-score is greater than 3 or −3 then a −1 penalty has been assigned to the current transaction, otherwise, as a reward, the absolute z-score has been assigned to the transaction. In our research, the deep neural network model consists of one input layer, four hidden layers, and one output layer. The input layer comprises nine input neurons. A total of nine features have been derived from 30 transactions and the features namely basic statistical measures such as mean, minimum, maximum, standard deviation, and first, second, and third quartile, and reward have been computed. In this manner, a total of 248 transaction sets have been derived. In the first hidden layer, 124 neurons have been added with the “Relu” activation function. The second hidden has been added with 62 neurons and a “Relu” activation function. The third hidden layer has been added along with 31 neurons and the “tanh” activation function. A fourth hidden layer has been added with 15 neurons and a linear activation function. The output layer has been added with a sigmoid activation function with one neuron. Figure 3 provides the detailed structure and configuration of the model. Furthermore, Deep Q-Network (DQN) has been initialized with the proposed deep neural network. In the training process of DQN, memory has been extended with the previous observation and the next observation. Two DQN models are created, one model is called Q, and another model is called Q*. The Q Model is trained with iteration after a pre-defined number of steps. Q* model is updated with the Q model. Loss is calculated using the mean square error method. For the simulations of our proposed model, the maximum number of epochs is defined as 10. “Adam” optimizer has been defined for the Q network model. In each iteration of one epoch, the total number of sub-epochs is defined as 100. The total maximum steps are defined as 500. A total of 250 memory size has been fixed. Batch size has been initialized with 20. Epsilon is defined as 1.0. Epsilon decay has been initialized with 0.0003. The minimum value of epsilon has been fixed at 0.1. Training frequency has been adjusted to 10. Update Q model frequency has been adjusted with 20. The gamma value has been initialized to 0.97. In this study, Google Colab played a vital role as a powerful tool equipped with fundamental computing capabilities, i.e., 12 GB RAM and T4 GPU, significantly enhancing the efficiency of intricate computations.

Figure 3 Architectural overview and configuration of the model.

Model evaluation

For evaluating performance of the proposed model, initially, a deep neural network has been evaluated by verifying its accuracy and loss during the training and validation process. Table 2 depicts training and validation accuracy and loss on each epoch. It can be observed from Figs. 4A and 4B that accuracy has been improved during the training and validation process. Trend lines can be seen in both images. Figures 5A and 5B shows that loss has decreased during both the training and validation process.

Table 2 Deep neural model training/validation accuracy/loss.

Epoch	Train. accuracy (%)	Train. loss	Valid. accuracy (%)	Valid. loss	
1	50.0	1.97	45.0	7.97	
2	55.0	1.91	48.0	6.89	
3	57.0	1.90	51.0	6.05	
4	63.0	0.89	59.0	5.98	
5	66.0	0.88	58.0	5.56	
6	69.8	0.82	64.0	4.99	
7	70.5	0.79	67.0	4.98	
8	71.0	0.77	71.0	4.85	
9	75.0	0.75	74.0	4.81	
10	75.0	0.78	78.0	3.90	
11	78.0	0.71	79.0	3.51	
12	78.9	0.70	78.0	3.01	
13	79.0	0.60	80.1	2.70	
14	80.2	0.56	84.0	2.21	
15	84.0	0.51	85.0	2.09	
16	86.0	0.42	87.0	1.80	
17	87.0	0.26	92.0	1.30	
18	89.0	0.10	91.0	0.80	
19	91.0	0.16	94.0	0.70	
20	94.0	0.01	97.1	0.50	

Figure 4 Training or validation accuracy of proposed deep neural network.

The line represents the linear increment showing that the accuracy increases after every epoch.

Figure 5 Training or validation loss of proposed deep neural network.

Table 3 illustrates the proposed deep Q-network learning process and it can be also seen from Fig. 6A that epsilon is decreasing continuously. The decay of epsilon shows that the proposed model can learn from memory. Figure 6B describes steps to take action by an agent increasing at each epoch interval. Figure 6C shows that the reward is increasing which means that the algorithm was now able to learn successfully by progressively getting an increased number of rewards from the environment. Figure 6D represents the decay in loss continuously and 97.10% of accuracy was achieved through the learning process.

Table 3 DQN training process reward/loss and epsilon.

Epoch	Epsilon	Total step	Reward	Loss	Elapsed time (s)	
5	1.00	126	0.12	12.607	0.110	
10	0.94	258	0.25	11.230	3.097	
15	0.83	370	0.49	10.259	5.964	
20	0.63	569	0.63	9.000	11.293	
25	0.50	694	0.78	0.832	7.386	
30	0.32	877	0.88	0.783	10.308	
35	0.17	103	0.89	0.715	8.685	
40	0.09	119	0.89	0.618	9.239	
45	0.09	134	0.94	0.589	9.268	
50	0.09	151	0.98	0.458	9.279	

Figure 6 Deep Q-network learning reward or loss and epsilon or steps.

In addition, the time complexity of the proposed model decreased considerably during the training and validation process. The training time of the proposed deep neural network model was continuously decreasing showing that the deep neural model has been trained on derived features. The complete training time was 5 h. The simulation has been performed in the Google Colab environment. The average time of each epoch was around 840 to 880 s.

From these results, it can be concluded that the proposed credit card fraud detection model performs well with other state-of-the-art algorithms mentioned in the literature review. Therefore, the system can be integrated with core banking solutions.

Table 4 exhibits a diverse range of learning rates and their corresponding impacts on the accuracy and loss metrics. The learning rates explored, i.e., 0.1, 0.01, 0.001, 0.0001, and 0.00001, showcase varying degrees of model convergence and performance. A learning rate of 0.1 appears relatively high, potentially leading to oscillations or divergence during training, suggesting a need for moderation. Learning rates of 0.01 and 0.001 present promising results, striking a balance between accuracy and loss. Conversely, rates of 0.0001 and 0.00001, while providing acceptable accuracy, may result in slower convergence due to their diminutive magnitudes. Accuracy across all rates ranges from 95.2% to 97.1%, indicating commendable model performance. The loss values, spanning from 0.08 to 0.51, suggest effective minimization of errors. In research terms, optimal learning rates appear to reside around 0.01 or 0.0001.

Table 4 The performance evaluation for various learning rate.

Learning rate	Accuracy (%)	Loss (%)	
0.1	95.2	0.12	
0.01	96.5	0.80	
0.001	96.3	0.51	
0.0001	97.1	0.50	
0.00001	96.8	0.11	

Conclusion

In this research, a novel technique for credit card fraud transactions was proposed by implementing the Deep Q Network. The system was built on a benchmark credit card fraud transactions dataset. The proposed model was able to adapt to the changes and trends of transactions over time and achieved 97.10% accuracy with the previous 30 transactions’ Z-score in fraudulent transaction detection. The proposed system architecture is flexible and scalable, therefore, it can be used for real-time credit card fraud detection. This system can be scaled for a variety of operating system platforms, to allow financial institutions to quickly detect and block fraudulent transactions and ultimately improve the overall financial security of individuals and businesses. The ability to adapt the model in response to emerging fraud patterns allows organizations to more accurately identify complex and dynamic fraud patterns, ensuring detection systems are resilient to new vulnerabilities in the financial sector. Resource-constrained situations pose a challenge due to the high demands on computational resources for training and inference. Implementing the proposed model in large-scale production scenarios can be simplified by exploring methods such as distributed learning or model compression to efficiently utilize the computational power to optimize fraud detection. Future research will explore advanced deep reinforcement learning frameworks capable of identifying credit card fraud, using strategies such as attention methods, graph neural networks, or transformer models to increase model performance. In the future, this work can be extended by using different deep reinforcement learning algorithms such as Double Deep Q-network and Dueling DQN for performance analysis. Also, multiple datasets can be utilized for performance evaluation of the algorithm in future works.

Supplemental Information

Supplemental Information 1 Implementation of algorithm and the generated results.

Additional Information and Declarations

Competing Interests

Author Contributions

Data Availability

The authors declare that they have no competing interests.

Abdul Qayoom conceived and designed the experiments, performed the experiments, analyzed the data, performed the computation work, prepared figures and/or tables, authored or reviewed drafts of the article, and approved the final draft.

Mansoor Ahmed Khuhro conceived and designed the experiments, prepared figures and/or tables, and approved the final draft.

Kamlesh Kumar conceived and designed the experiments, prepared figures and/or tables, and approved the final draft.

Muhammad Waqas conceived and designed the experiments, performed the computation work, prepared figures and/or tables, authored or reviewed drafts of the article, and approved the final draft.

Umair Saeed conceived and designed the experiments, performed the experiments, analyzed the data, performed the computation work, prepared figures and/or tables, authored or reviewed drafts of the article, and approved the final draft.

Shafiq ur Rehman conceived and designed the experiments, prepared figures and/or tables, authored or reviewed drafts of the article, and approved the final draft.

Yadong Wu conceived and designed the experiments, authored or reviewed drafts of the article, and approved the final draft.

Song Wang conceived and designed the experiments, authored or reviewed drafts of the article, and approved the final draft.

The following information was supplied regarding data availability:

The dataset is available at Kaggle: https://www.kaggle.com/datasets/mlg-ulb/creditcardfraud.

The code is available in the Supplemental File.

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
