# Peer review of "A novel approach for credit card fraud transaction detection using deep reinforcement learning scheme"

_PeerJ Computer Science, doi:10.7717/peerj-cs.1998_

## Round 0.1 · original submission · Major Revisions

I have received reviews of your manuscript from scholars who are experts on the cited topic. They find the topic very interesting; however, several concerns must be addressed regarding experimental results (more datasets), comparisons with current approaches, and references. These issues require a major revision. Please refer to the reviewers’ comments listed at the end of this letter, and you will see that they are advising that you revise your manuscript. If you are prepared to undertake the work required, I would be pleased to reconsider my decision. Please submit a list of changes or a rebuttal against each point that is being raised when you submit your revised manuscript.

Thank you for considering PeerJ Computer Science for the publication of your research.

With kind regards,

·

Basic reporting

Clear and unambiguous, professional English language is used throughout the paper. The literature references does not provide sufficient field and background context. More specific references supplying data on the number of credit card fraud/attacks, characteristics of individuals engaging in fraud/attacks, and how much they cost individuals and businesses, and the efficacy of the state of the art techniques needs to be better established. Specific references that address these concerns are below. The information with them should be summarized and they should be referenced to establish the scale and extent of the problem of credit card fraud.

Barker, Katherine J., Jackie D'amato, and Paul Sheridon. "Credit card fraud: awareness and prevention." Journal of financial crime 15.4 (2008): 398-410.

Dheepa, V., and R. Dhanapal. "Analysis of credit card fraud detection methods." International journal of recent trends in engineering 2.3 (2009): 126.

Beigi, S., and M. R. Amin Naseri. "Credit card fraud detection using data mining and statistical methods." Journal of AI and Data Mining 8.2 (2020): 149-160.

Bolton, Richard J., and David J. Hand. "Statistical fraud detection: A review." Statistical science 17.3 (2002): 235-255.

Daniele Vernon-Bido et al. 2016. Towards modeling factors that enable an attacker. In Proceedings of the Summer Computer Simulation Conference (SCSC '16). Society for Computer Simulation International, San Diego, CA, USA, Article 46, 1–6.

Shadel, Doug, and Karla Blair Schweitzer Pak. The psychology of consumer fraud. Diss. Universiteit van Tilburg, 2007.

Williams, Albert A., April Lewis-Parks, and William Wolf. "Do Demographic, Psychological, and Financial Characteristics Increase the Likelihood to be Victims of Credit Card Fraud?." Journal of Personal Finance 22.2 (2023).

Self-contained with relevant results to hypotheses.

The article structure of the manuscript is professional. However, the professionalism of the figures and tables could be improved. Figures 1 is blurry. Figure 3 is stretched and not at an appropriate height/width ratio. In additon, the caption does not make it clear what line represents in Figure 4 (top and bottom). The numbers in the tables in the paper are difficult to compare to one another because the numbers are not right justified. As a result, the significant digits of the numbers are not aligned on top of one another (i.e. 100ths, 10ths, etc.) Right justifying the numbers in the tables will improve readability and enable readers to compare results between rows.

A data file only featuring the raw data is not shared. Instead, the data is embedded in a python notebook. The data should be broken out seperate from the code and be shared.

The results are self-contained and relevant to the hypotheses. The formal results include clear definitions of all terms and theorems. The work presents original primary research that is within the scope of the journal.

Experimental design

The research question is well-defined. However, appropriate context of the state of the art is not provided so it is not clear to what extent it is relevant and meaningful. Along with the issues identified previously related to the scope and context of the research question, it is not clear how well existing state of the art techniques would perform on the same evaluation data as the technique proposed by the authors. Without comparing the proposed technique to others on a testbed it is unclear to the extent the research fills an identified gap. Related to this the investigation that compares the proposed work to other state of the art techniques needs to be rigorous. Specifically, the proposed technique should be shown to be statistically significantly more effective at detecting credit card fraud compared to other state of the art techniques. The python notebook provided helps ensure the methods are described with sufficient detail and information to replicate it. However, a README file discussing the structure of the notebook and additional comments in the notebook would improve the ability to replicate.

Validity of the findings

As mentioned previously the impact and novelty of the solution are not assessed because their is not a sufficiently robust comparison to other state of the art techniques on the same testbed for evaluation. Doing this and testing for statistically significant improvements from the proposed approach with respect to the state of the art techniques would address this concern. Statistically sound and controlled evaluation data related to these comparisons have not been provided, they need to be. The conclusions are well stated and linked to the original research questions.

Additional comments

There is a distinct lack of motivation and context that needs to be addressed in this paper. In addition, there needs to be a comparative evaluation of the proposed approach against other state of the art techniques where the performance of the proposed approach is tested for statistical significance with respect to the current state of the art practices.

Reviewer 2 ·

Basic reporting

In this research, a novel technique for credit card fraud transactions was proposed through the implementation of the Deep Q Network. The novelty is minimum.

Experimental design

No comment.

Validity of the findings

More experimental results required.

Additional comments

What is the purpose of using deep Q network? Compare the performance of deep Q network with other deep learning models.
What happened while varying the learning rate. The performance evaluation for various learning rate is required.
Write the motivation in the Introduction section.
The research gaps identified in the existing methods should be presented.
Consider more datasets for the performance evaluation.
The parameter details must be tabulated.
Compare the performance of the proposed method with other methods in the literature review.
The applications, limitations, and future work of the model should be mentioned in the Conclusion section.

Reviewer 3 ·

Basic reporting

• The basic reporting of the manuscript is clear. The motivation could be enhanced. Here I fail to see the novelty or the added value of the approach apart from using Deep Q network for credit card fraud detection. There are still some more profound works that are not reviewed in this work. Research gaps are missing in this manuscript and authors should have claimed the contribution after highlighting the research issues of existing techniques. The paper should give depth analysis.

Experimental design

• The model design section presents only the theoretical explanation of the Deep Q network and lacks novelty. The experimental setup can be improved. The proposed method is not compared with the state-of-the-art/related works in terms of experimentation, results, and suitability. The dataset used in this research has been taken from the Kaggle dataset repository. It consists of credit card transactions of European card holders. It is also suggested to work on the different categories of datasets that consist of different features.

Validity of the findings

It looks that some significant findings were observed using Deep Q network. However, the findings can be evaluated after clarifying the models.

Additional comments

More recent papers should be reviewed. There are still some more profound works that are not considered in this work. The writing of this manuscript needs further improvements. There are some grammatical mistakes.

---

## Round 0.2 · accepted · Accept

I am pleased to inform you that your work has now been accepted for publication in PeerJ Computer Science.

Please be advised that you are not permitted to add or remove authors or references post-acceptance, regardless of the reviewers' request(s).

Thank you for submitting your work to this journal. On behalf of the Editors of PeerJ Computer Science, we look forward to your continued contributions to the Journal.

With kind regards,

·

Basic reporting

My concerns have been sufficiently addressed by the revisions. The paper is now suitable for publication. It uses clear and unambiguous, professional English throughout. The literature references, provide sufficient field background/context. The paper has professional article structure, figures, tables. The raw data is shared. The paper is self-contained with relevant results to hypotheses. The paper provides formal results which include clear definitions of all terms and theorems, and detailed proofs.

Experimental design

My concerns have been sufficiently addressed by the revisions. The paper is now suitable for publication. The paper presents original primary research within the aims and scope of the journal. It's research question is well defined, relevant and meaningful. It states how the research fills an identified gap. The paper provides a rigorous investigation performed to a high technical and ethical standard. The methods in the paper are described with sufficient detail and information to replicate.

Validity of the findings

My concerns have been sufficiently addressed by the revisions. The paper is now suitable for publication. The paper provides sufficient detail for meaningful replication. The rationale and benefit to the research literature is clearly stated. All underlying data have been provided. The data are robust, statistically sound and controlled. The conclusion is well stated and linked to the original research question. The conclusions are limited to supporting the presented results.